# Effect of Evolution of Carbon Structure during Torrefaction in Woody Biomass on Thermal Degradation

**DOI:** 10.3390/ijerph192416831

**Published:** 2022-12-15

**Authors:** Peng Liu, Panpan Lang, Ailing Lu, Yanling Li, Xueqin Li, Tanglei Sun, Yantao Yang, Hui Li, Tingzhou Lei

**Affiliations:** 1National-Local Joint Engineering Research Center of Biomass Refining and High-Quality Utilization, Institute of Urban and Rural Mining, Changzhou University, Changzhou 213164, China; 2Changzhou Key Laboratory of Biomass Green-Safe & High Value Utilization Technology, Changzhou 213164, China; 3State Key Laboratory of Utilization of Woody Oil Resource, Hunan Academy of Forestry, Changsha 410004, China

**Keywords:** torrefaction, thermal degradation, carbon structure, biomass, oxygen functional group, DTG

## Abstract

Torrefaction is an effective method for upgrading biomass. Cedar torrefaction is carried out in a fixed bed reactor at the temperature of 200–300 °C. The structural parameters are obtained from elemental analysis and ^13^C nuclear magnetic resonance (NMR). Thermal degradation behavior of raw and torrefied cedar is monitored by thermogravimetry analysis. The results show that carbon structure varied during torrefaction has a significant effect on thermal degradation of cedar. Some unstable oxygen functional groups, such as C1 of hemicellulose, β-O-4 linked bonds, and amorphous C6 of cellulose, are decomposed at mild torrefaction of torrefied temperature ≤ 200 °C. The temperature of maximum weight loss rate increases from 348 °C of raw cedar to 373 °C of C-200. The amorphous cellulose is partly re-crystallized at moderate torrefaction of torrefied temperature 200–250 °C. The aromaticity of torrefied cedar increases from 0.45 of C-200 to 0.73 of C-250. The covalent bond in the side chain of aromatic rings in cedar was further broken during torrefaction at severe torrefaction of torrefied temperature 250–300 °C. The area percentage of DTG mainly signed at 387 °C of C-300. The proton aromatic carbon increases from 12.35% of C-250 to 21.69% of C-300. These results will further facilitate the utilization of biomass for replacing fossil fuel to drive carbon neutrality.

## 1. Introduction

Biomass is the only renewable energy with carbon resource that can be an alternative to fossil fuel for producing chemicals and materials [1,2]. Meanwhile, biomass conversion is carbon neutral during its lifecycle utilization, which can realize cycle utilization of carbon element [3]. Biomass thermal conversion to chemicals and materials is exhaustive and low cost, but it is challenging due to some unfavorable features of biomass [4], such as low bulk density, high oxygen content, and weak grindability. The high oxygen content leads to high water content, strong hydrophilicity, and low energy density. These unfavorable features have limited the wide application of biomass. It is necessary to develop effective methods for upgrading biomass aims to prevent these features.

Torrefaction is considered as a widely used technology for optimizing the fuel quality [5], pyrolysis behavior [6], gasification performance [7], and combustion characteristics [8].

Compared with raw biomass, torrefied biomass has a higher calorific value with higher energy density due to increased carbon contents and decreased oxygen contents. Torrefied biomass with easier grindability and hydrophobicity can reduce the cost of storage and transformation. The pyrolysis behavior of torrefied biomass is similar to high-volatile coal [9]. Torrefaction is an effective method for promoting bio-oil and pyrolysis gas quality [10]. Pretreatment by torrefaction is an efficient strategy for enhancing syngas production whilst minimizing tar formation. The emissions of C=O/C-O compounds is reduced and the release of CO_2_, CO, and H_2_O is enhanced during combustion pretreated by torrefaction [11]. Integration of torrefaction and pelletization decreased the emissions of NO and SO_2_ during combustion of biomass, and the combustion slags also exhibited favorable potentials for soil regulation [12].

Torrefaction is carried out at the temperature range of 200–300 °C. In that temperature range, hemicelluloses decompose but there is only slight decomposition of cellulose and lignin performed on account of inherent difference in the structures of the three components [13]. The structure of biomass is mainly influenced by the temperature, atmosphere, and duration during torrefaction [14]. Some studies investigated the structural variation during biomass torrefaction using thermogravimetric analysis (TG), scanning electron microscopy (SEM), Fourier transform infrared analysis (FTIR), Brunauer–Emmett–Teller (BET), X-ray photoelectron spectroscopy (XPS), and nuclear magnetic resonance (NMR) [15]. Chen [16] suggested the distributions of TGA and DTG providing a qualitative insight into the relationship among hemicellulose, cellulose, and lignin indicated light torrefaction for the depletion of hemicellulose and slight impact on cellulose and lignin. The authors of [17] characterized the properties and structures of the raw and torrefied biomass wastes extensively by proximate, elemental, fiber, calorific, thermogravimetric, and SEM and FTIR analyses showing that torrefied biomasses approach high-volatile coal when the torrefaction temperature and duration increased. Zhang [18] investigated the effects of torrefaction temperature, atmosphere, and duration on the functional group distribution via FTIR, Raman spectra, and XPS, which showed that oxidizing agents could accelerate the conversion of surface hydrophilic groups, strengthening the hydrophobicity, and tend to promote the cracking of condensable aromatic structures with an increasing proportion of active structures in the torrefied rice husk. Melkior [19] showed light different transformations of the biomass components: de-acetylation of hemicelluloses, demethoxylation of lignin, changed in the cellulose structures during torrefaction by NMR and EPR. Zheng [20] used FTIR and quantitative solid ^13^C-NMR analysis of torrefied corncobs to show that the devolatilization, crosslinking, and charring of corncobs during torrefaction could be responsible for the bio-oil yield penalties. Previous research showed structural variation in biomass components influenced by temperature, atmosphere, and duration during torrefaction. Oxidative torrefaction [14,21] promotes the reaction rate of torrefaction, resulting in depolymerization and degradation of macromolecules in biomass, shortening the torrefaction duration, which could achieve a circular economy using flue gas as the oxidative torrefaction medium [22]. Torrefaction temperature [23,24] is a vital factor of torrefaction severity, stronger than duration. Though the effect of torrefaction on overall carbon structure could be quantifiably monitored by solid ^13^C-NMR and the presence of functional groups could be analyzed by FTIR to predict subsequent conversion behavior [25,26]. The quantitative information on structural evolution of biomass torrefaction is still insufficient. These structural variations have significant effect on cleavage of bond in biomass during thermal degradation. A comprehensive insight into mechanism on biomass microstructural evolution to predict thermal degradation after torrefaction is also indistinct.

The objective of this study was to obtain detailed structural information available to infer mechanism among biomass microstructural evolution at the temperature range of 200–300 °C during torrefaction monitored by solid ^13^C-NMR with a peak-fitting method. Thermal degradation was analyzed using DTG curves for further verifying the bond fracture properties with the increasing temperature. These results will guide industrial preparation of biomass solid fuel from wood wastes.

## 2. Experimental Sections

### 2.1. Material Preparation and Torrefaction

A representative wood waste, cedar, was selected from southeast of Anhui province in China. It was triturated to <0.2 mm and stored in desiccator. The proximate analysis was monitored according to standard methods shown in our previous paper [27]: moisture content (UNE-EN 14774-1:2010), ash content (UNE-EN 14775:2010), volatile matter (UNE-EN 15148:2010), and fixed carbon (determined by mass difference). The elemental analyzer (Elementar Vario Micro Cube, Elementar Analysensysteme GmbH) was used to analyze elements of C, H, N, S, and the O element was calculated by mass difference. Cellulose, hemicellulose, and lignin contents were determined by classical method proposed by Van Soest. These analyses of wood wastes are shown in Table 1.

### 2.2. Torrefaction

Torrefaction of cedar was carried out in a vertical, 800 mm × 40 mm diameter quartz tubular reactor. The samples were placed in the middle 200 mm of the reactor, where temperature was most uniform. An amount of 5 g of cedar was placed in the reactor and heated at 10 °C/min to a setpoint of 200, 225, 250, 275, or 300 °C for 30 min. An exhaust tube was connected to the bottom of the reactor with three levels of ice bath condensation. N_2_ was injected into the reactor at a flow rate of 90 mL/min to create an inert environment and remove the high temperature torrefied gas. The torrefied cedar samples were designated, per their process temperatures, as C-200, C-225, C-250, C-275, and C-300.

### 2.3. Analysis

^13^C NMR experiments of raw and torrefied cedar were performed on an Agilent 600 DD2 spectrometer at a resonance frequency of 150.15 MHz. ^13^C NMR spectra were recorded with spinning rate of 15 kHz with a 4 mm probe at room temperature using a solid cross polarization/magic angle spinning (CP/MAS) probe, the total suppression of sidebands sequence (TOSS). The measurement conditions were as follows: the contact time was 1 ms, with a recycle delay of 5 s and the scan number was 2048.

TG of raw and torrefied cedar is analyzed using a Pyris 1 TGA from Perkin Elmer, Waltham, MA, USA. Samples of 15 mg were heated from ambient temperature to 100 °C for 1 h to remove free water, then from 100 °C to torrefied temperatures keeping 30 min by heating rate of 10 °C/min to obtain torrefied cedar. Finally, samples were heated from the torrefaction setpoint temperature to 900 °C by heating rates of 10 °C/min to calculate kinetic parameters, with carrier gas by a 99.999% helium (He) flowing at 20 mL/min for an inert environment. The temperature corresponding to the maximum weight loss rate of the samples were determined from the first derivative of the weight loss curve with respect to temperature (DTG). The peak clusters in DTG curves contained various bond cleavage in raw and torrefied cedar. For interpreting the detailed data, the peak separation and quantitative calculation were performed using a curve-fitting program of Origin 9.1 by gaussian method.

## 3. Results and Discussion

### 3.1. Effect of Torrefaction on Proximate and Ultimate Analysis of Cedar

Table 2 shows the effect of proximate and ultimate analysis during torrefaction from cedar at the temperature range of 200–300 °C. Temperature is the main factor for variation of chemical composition compared with time. The volatile decreases from 80.04% of cedar to 36.71% of C-300 with temperature increasing during torrefaction and ash contents increase. That is attributed to enhance devolatilization during torrefaction. The ultimate analysis shows that the carbon increases with temperature increasing. The analyses of chemical element show maturity of biomass, especially the hydrogen index (H/C atomic ratio) and oxygen index (O/C atomic ratio). The Van Krevelen diagram shown the hydrogen index as a function of oxygen index plotted in Figure 1.

Upon the torrefaction process, both hydrogen index and oxygen index decrease. The hydrophilic ability of biomass decreases as the result of decrease in O/C. The chemical stability is higher when both indexes decrease. The torrefied cedar is expected to be more resistant to degradation. The fuel with low H/C and low O/C, left region of the plot, correspond to high maturity fuel. The other index assessed the maturity of the biomass, aromaticity, could be calculated by traditional empirical formula according to V_daf_ and C_daf_ proposed by Van Krevelen [28] shown in Equation (1). The variation of aromaticity during torrefaction is shown in Figure 2.
f_a_ = (100 − V_daf_) × 1200/1240 × C_daf_(1)

Seen from Figure 2, the aromaticity increases from 34.79% of cedar to 74.20% of C-300. That results from the weak bond cleavage in aliphatic hydrocarbons in hemicellulose and cellulose. The aromatic rings in lignin in cedar is mainly remained during torrefaction. The structural evolution in cedar has an essential change torrefied bellow 250 °C as aromaticity increasing obviously.

### 3.2. Detailed Evolution of Carbon Structure during Torrefaction from Cedar

The detailed structural evolution is monitored by ^13^C-NMR. The fitting curves in ^13^C-NMR spectra are shown in Figure 3a–d.

From Figure 3, the spectra of raw cedar and C-200 are almost the unchanged after thermal treatment at 200 °C. When the temperature increases to 250 °C, the intensity of signals up to 100 ppm becomes stronger than that of cedar and C-200. The intensity of signals bellow 100 ppm decreases obviously after torrefaction at 300 °C. The signals in ^13^C-NMR spectra corresponding to all the different carbons of the main wood components rely on the results of many our previous studies [27] shown in Table 3.

The quantitative analyses of the NMR spectra by curve-fitting method shown in Table 3. The compositions in cedar have a variation during torrefaction. For cellulose and hemicellulose, the focus was set on resonances arising from the carbon atoms as signal 6 assigning to C1 in cellulose, signal 7 assigning to C1 in hemicellulose, signal 9 assigning to amorphous C4 of cellulose, signal 10 assigning to C2,3,5 in hemicellulose and cellulose or Cα-OH in β-O-4-linked side chains, signal 12 assigning to amorphous C6 of cellulose, and signal 1 assigning to carbonyl carbon. Hemicellulose is sensitive to torrefaction. The representative signal 7 decreases to 0 after torrefaction over 200 °C. The C2,3,5 in hemicellulose and cellulose also decomposes from 36.18% of cedar to 7.25% of C-300. The amorphous cellulose is degraded as the signal 9 and 12 decreasing. The crystalline cellulose has a little initial increase during torrefaction bellow 250 °C. The crystalline C4 in cellulose is more stable than crystalline C6 in cellulose. The crystalline C4 in cellulose start to cleave at 250 °C and the crystalline C6 in cellulose at 200 °C. These demonstrate the amorphous cellulose is partly re-crystallized during torrefaction at the low temperature range, then degraded at a higher temperature. These unstable oxygen functional groups decompose leading to lower O/C and H/C ratio shown in Figure 1.

For lignin, the main degradation is the cleavage of side chains and demethoxylation. The proton aromatic carbon in lignin assigned at 122 ppm and nonetherified syringyl unit assigned at 132 ppm increase remarkably, indicating that the demethoxylation is dominant reaction during torrefaction. Demethoxylation of syringyls is expected to lead to an increase in guaiacyls at a low temperature range for torrefaction which contribute to area variation of signal 2 and 3 [29]. Above 250 °C, the curve of signal 2 and 3 reaches a plateau that may indicate that demethoxylation begins to occur in guaiacyl units. The aromaticity in torrefied cedar increases shown in Figure 2. However, the carbonyl carbon assigned at 178 ppm increases caused by the stable oxygen functional group remaining during torrefaction.

### 3.3. Effect of Structural Evolution on Thermal Degradation by DTG

Biomass pyrolysis, comprised of radical generation by cleavage of covalent bond, generates free radicals and product formation by radical fragments coupling [30,31]. Biomass, composed by lignin, cellulose, and hemicellulose, has a number of covalent bonds which is influenced by structure, including C_al_-C_al_, C_ar_-C_ar_, C_ar_-C_al_, C_ar_-O, C_al_=C_al_, C_al_-O, and C_al_=O bonds [32]. The cleavage of these bonds, correlated with the DTG curves, are results in difference on thermal degradation during biomass pyrolysis. The DTG curves of raw and torrefied cedar are shown in Figure 4a–d.

Figure 4 shows the cleavage of covalent bond according to DTG analysis, which is mainly caused by decomposition of carbonyl groups, breakage of C-O-C, C_al_-C_al_ in hydro-carbonates and side chains of aromatics in lignin during primary pyrolysis. The stable C_ar_-C_ar_, and C_al_=O bonds may be broken at a higher pyrolysis temperature. The DTG curves in Figure 4 show the overlapped signals for raw and torrefied cedar during thermal degradation. The signals of raw cedar are fitted by five sub-curves, four of torrefied cedar at 300 °C and seven of torrefied cedar at 200–250 °C representing major covalent bonds cleavage. Peak 1 is correlated to the decomposition of carbonyl groups, hydrogen bonds, and weak C-O bonds, at the temperature range of 235–270 °C [33]. Peak 2 is dominated by cleavage of C-O bonds at the temperature range of 290–315 °C. The weak C_al_-C_al_ bonds with bond energy range of 295–325 kJ/mol [32] is assigned at peak 3 shown at about 350 °C. This pyrolysis stage is decomposition of cellulose and maximum weight loss rate of raw cedar located in this peak. Peak 4 is correlated to the breakage of C_al_-C_al_ bonds at about 370–390 °C, associated with cellulose pyrolysis and weak aromatic side chain breakage. Peak 5 can be assigned to stable oxygen functional group such as C=O conjugated to aromatic rings at 445–510 °C. Peak 6 is the breakage of strong aromatic side chains and Peak 7 represents condensation of aromatic rings. The parameters of curve fitting operation, i.e., peak temperature (T_p_) and peak area (A_p_), calculated from Figure 4a–d are listed in Table 4.

Seen from data in Table 4, the decomposition of unstable oxygen functional groups is lower after torrefaction according to fitting results of DTG. The temperature of maximum weight loss rate (T_m_) increases with the torrefied temperature. Decomposition of unstable oxygen functional groups in hemicellulose leads to peak 1 and 2 of DTG fitting curves. The area percentage of peak 2 decreases from 29.94% of raw cedar to 0 of C-300 consistent with signal 7 and 10 decreasing. Peak 3 and 4 is the main cellulose decomposition. The unstable amorphous cellulose is decomposed preferentially. The amorphous cellulose in raw cedar is highest and oxygen carbon in raw cedar is higher than other torrefied cedar so that the maximum weight loss rate signed in peak 3 and T_m_ is the lowest. With the torrefaction processing, the amorphous cellulose is partly re-crystallized according to peak area at T_m_. The T_m_ increases to 387 °C of C-250. Peaks 5, 6, and 7 indicate the thermal degradation of aromatic rings in lignin. The area percentage increases with the increasing temperature of torrefaction due to enhancement of aromatic rings in lignin during torrefaction.

### 3.4. Proposed Mechanism on Torrefaction of Cedar

According to the gradient on the dot in Van Krevelen diagram, torrefaction of cedar can divide into three steps in the temperature range of 200–300 °C, shown in Figure 5.

First, with the torrefied temperature ≤ 200 °C, called mild torrefaction, the H/C decreases from 1.62 to 1.23 and O/C from 0.49 to 0.42. A little unstable oxygen functional groups with weak bond energy, such as C1 of hemicellulose, β-O-4 linked bonds, and amorphous C6 of cellulose, are decomposed to form CO_2_. Hydrogen bonds in cedar have been rearranged. The area percentage of DTG at Peak 2 decreases from 29.94% to 18.48%. However, the carboxyl carbon in hemicellulose increasing from 3.49% to 9.66% indicates that mild torrefaction has a little rearrangement of bond in hemicellulose. The area percentage of DTG at Peak 1 changes a little. The T_m_ increases from 348 °C to 373 °C.

Second, at a torrefied temperature 200–250 °C, called moderate torrefaction, the H/C decreases from 1.23 to 0.87 and O/C from 0.42 to 0.19. The number of oxygen functional groups are decomposed unceasingly. The amorphous C4 and C6 of cellulose decreases. The crystalline C6 increases at 200 °C and crystalline C4 increases at 250 °C, then decreases at a higher torrefied temperature. The amorphous cellulose is partly re-crystallized during torrefaction at this stage. The aliphatic carbon increases with the oxygen functional group decomposition. The aromaticity of torrefied cedar increases from 0.45 to 0.73.

Third, at a torrefied temperature of 250–300 °C, called severe torrefaction, the H/C decreases from 0.87 to 0.76 and O/C from 0.19 to 0.10. The covalent bond further broken during torrefaction. The aliphatic carbon and crystalline cellulose decreases in cedar. The area percentage of DTG mainly signed at Peak 4. The proton aromatic carbon increases remarkably. The side chains of aromatic carbon in cedar decreases.

## 4. Conclusions

Torrefaction improves the energy quality of cedar. Structural evolution on torrefaction of cedar can divide into three steps in the temperature range of 200–300 °C, which has a significant effect on thermal degradation.

Above all, at a torrefied temperature ≤ 200 °C, called mild torrefaction, the O/C decreases a little. Part of the unstable oxygen functional groups, such as C1 of hemicellulose, β-O-4 linked bonds, and amorphous C6 of cellulose, are decomposed. The area percentage of DTG at Peak 2 decreases from 29.94 to 18.48%. The T_m_ increases from 348 to 373 °C.

Next, at a torrefied temperature 200–250 °C, called moderate torrefaction, the number of oxygen functional groups are decomposed unceasingly. The amorphous cellulose is partly re-crystallized during torrefaction. The aromaticity of torrefied cedar increases from 0.45 to 0.73.

Finally, at a torrefied temperature 250–300 °C, called severe torrefaction, the covalent bond in side chain of aromatic carbon in cedar further broken during torrefaction. The aliphatic carbon and crystalline in cedar cellulose decreases. The area percentage of DTG mainly signed at 387 °C. The proton aromatic carbon increases remarkably.

These results will further facilitate the utilization of biomass for replacing fossil fuel to drive carbon neutrality.

## Figures and Tables

**Figure 1 ijerph-19-16831-f001:**
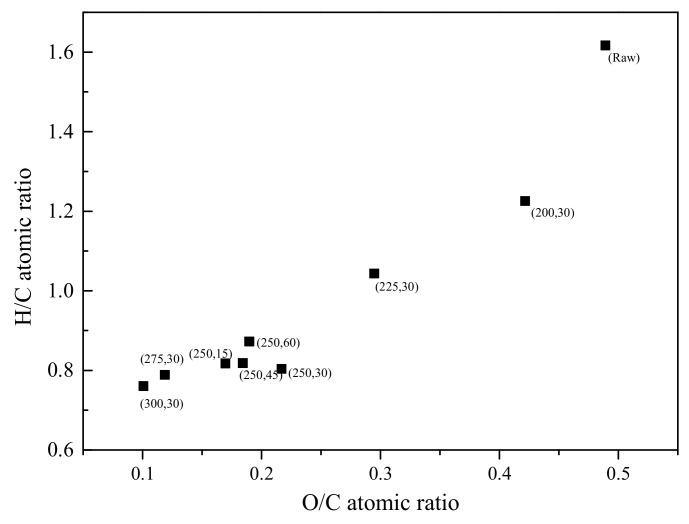
Van Krevelen diagram of torrefied cedar at temperature range of 200–300 °C.

**Figure 2 ijerph-19-16831-f002:**
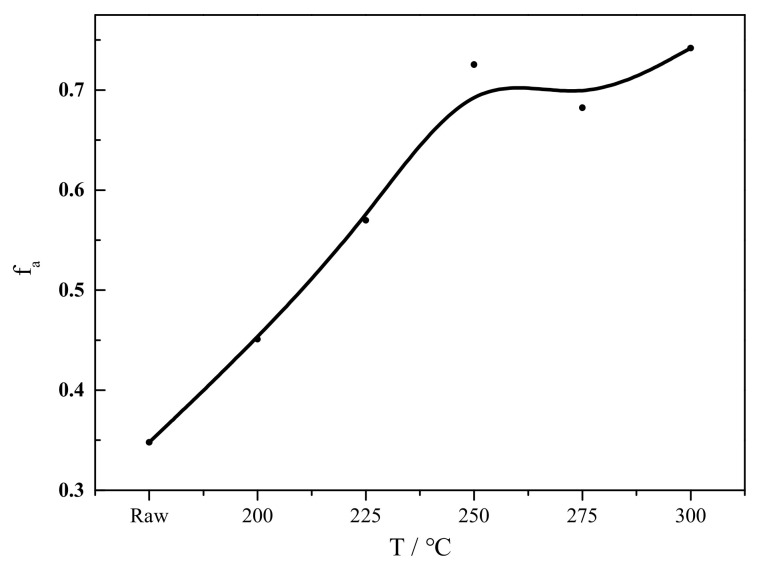
Variation of aromaticity with the torrefied temperature increasing.

**Figure 3 ijerph-19-16831-f003:**
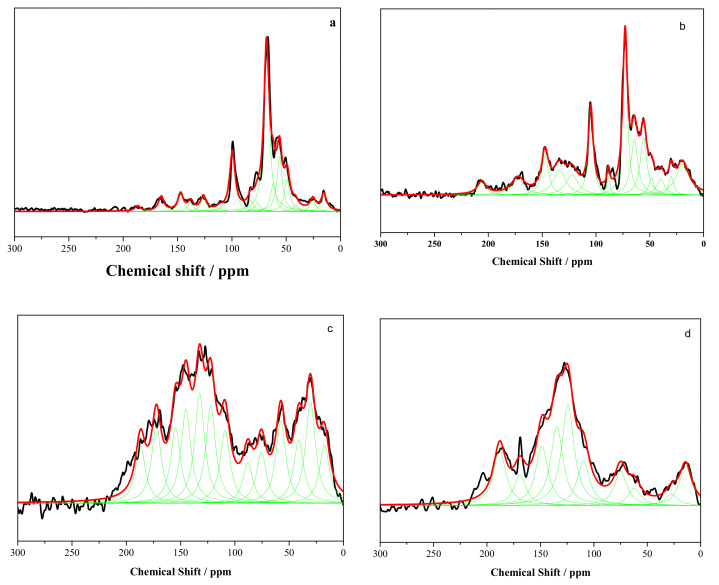
Fitting curves in ^13^C-NMR spectra of raw and terrified cedar ((**a**)—cedar, (**b**)—C-200, (**c**)—C-250, (**d**)—C-300). Green line: Fitting peak curve; Red line: Fitting curve; Black line: Raw curve.

**Figure 4 ijerph-19-16831-f004:**
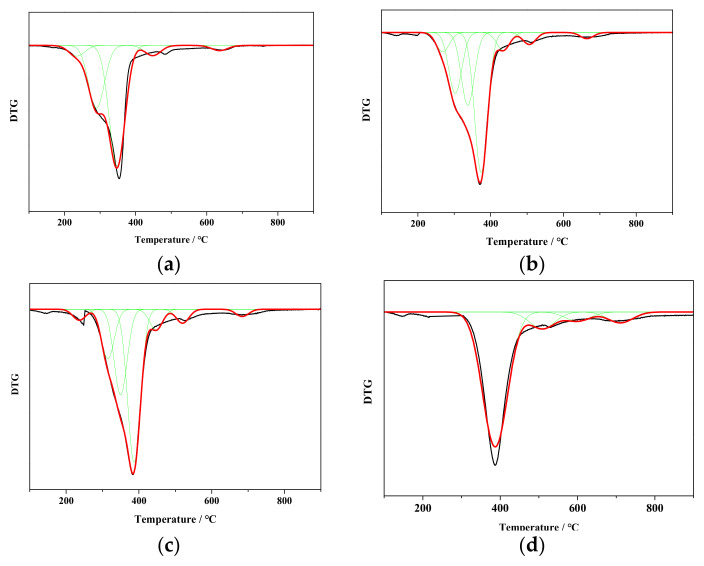
DTG fitting curves of raw and torrefied cedar. ((**a**)—cedar, (**b**)—C-200, (**c**)—C-250, (**d**)—C-300). Green line: Fitting peak curve; Red line: Fitting curve; Black line: Raw curve.

**Figure 5 ijerph-19-16831-f005:**
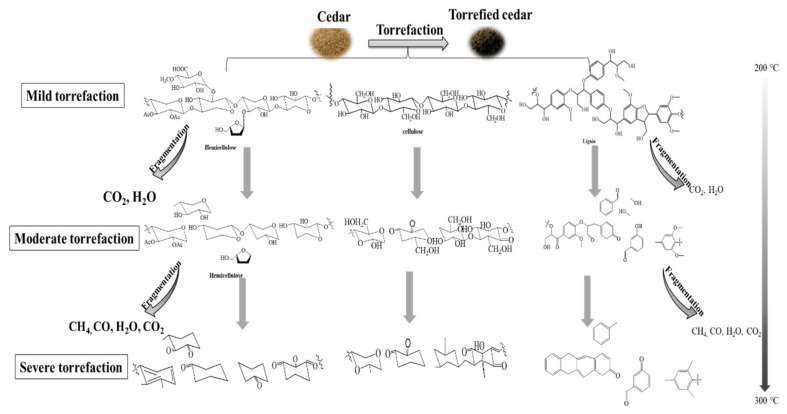
Structural evolution of cedar torrefaction.

**Table 1 ijerph-19-16831-t001:** Fundamental analysis of cedar.

Sample	Proximate Analysis, %			Ultimate Analysis, daf, %					Fiber Analysis, d, %			
	A	V_daf_	FC_daf_	C	H	N	S	O	Hemicellulose	Cellulose	Lignin	Others
Cedar	5.35	80.04	19.96	55.52	7.48	0.76	0.04	36.21	17.71	39.45	27.62	15.22

**Table 2 ijerph-19-16831-t002:** The proximate and ultimate analyses of raw and torrefied cedar.

Sample	Proximate Analyses %			Ultimate Analyses %				
	A_d_	V_daf_	FC_daf_	N_daf_	C_daf_	H_daf_	S_daf_	O_daf_ ^a^
Torrefied at a certain temperature for 60 min								
Cedar	4.79	80.04	19.96	0.76	55.52	7.48	0.04	36.21
C-200	7.11	72.21	27.79	0.75	59.63	6.09	0.01	33.52
C-225	8.46	60.52	39.48	0.78	67.05	5.83	0.00	26.35
C-250	10.82	44.02	55.98	1.00	74.68	5.43	0.00	18.89
C-275	13.15	43.04	56.96	1.11	80.79	5.31	0.00	12.79
C-300	13.72	36.71	63.29	1.12	82.55	5.23	0.00	11.09
Torrefied at 250 for a certain duration								
15	10.79	48.93	51.07	1.01	76.48	5.21	0.00	17.30
30	10.76	45.92	54.08	0.99	73.01	4.89	0.00	21.11
45	10.77	44.35	55.65	0.96	75.38	5.14	0.00	18.52
60	10.82	44.02	55.98	1.00	74.68	5.43	0.00	18.89

^a^ The contents of O element are caculated by difference.

**Table 3 ijerph-19-16831-t003:** Main carbon structure assigned at chemical shift with area percentage of raw and torrefied cedar.

No.	Chemical Shift/ppm	Assignments	Cedar	C-200	C-250	C-300
1	178	Carbohydrate; -**C**OO-R; CH_3_-**C**OO-	1.15	3.08	6.68	12.06
2	168	Lignin S^a^3(e^b^) S5€	3.20	5.60	8.56	6.48
3	149	Lignin G^c^1€ G4€ S3(ne^d^) S5(ne)	3.80	7.35	13.41	12.85
4	132	Lignin G1€ S1(ne), S4(ne)	2.07	8.04	11.21	16.96
5	122	Lignin G6, G5, S6, S2	3.15	6.61	12.35	21.69
6	110	Cellulose C1	0.62	-	8.29	9.45
7	101	Hemicellulose C1	12.95	10.87	-	-
8	84	Crystalline C4 of cellulose	2.46	1.45	3.52	-
9	80	Amorphous C4 of cellulose, lignin C_β_	1.91	-	-	-
10	70	Hemicellulose and cellulose C2,3,5 Cα-OH inβ-O-4-linked side chains	36.18	18.90	5.12	7.25
11	63	Crystalline C6 of cellulose	6.11	10.33	2.63	-
12	59	Amorphous C6 of cellulose; Lignin C_γ_	12.00	9.94	6.91	3.69
13	50	Lignin OCH_3_	7.04	2.38	-	-
14	43	Aliphatic C-C	1.42	3.32	6.09	-
15	26	Aliphatic CH_2_	2.40	2.47	10.02	2.33
16	15	Hemicellulose CH_3_-COO-	3.49	9.66	5.21	7.24

Note: ^a^, S is syringyl unit; ^b^, e is etherified; ^c^, G is guaiacyl units; ^d^, ne is nonetherified.

**Table 4 ijerph-19-16831-t004:** Fitting results of DTG curves.

Peak	DTG-Cedar		DTG-C-200		DTG-C-250		DTG-C-300	
	T_p_/°C	A_p_/%	T_p_/°C	A_p_/%	T_p_/°C	A_p_/%	T_p_/°C	A_p_/%
1	235	5.09	268	5.89	236	3.23	-	-
2	290	29.94	302	18.48	314	14.53	-	-
3	348	57.61	337	22.14	349	25.09	-	-
4	-	-	373	42.56	387	44.88	387	78.61
5	448	4.87	434	5.42	447	6.07	508	9.74
6	-	-	506	3.64	520	4.08	600	5.31
7	636	2.49	664	1.86	684	2.13	710	6.34

## Data Availability

Raw data of this article are available upon request to corresponding author.

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
