# Peer review of "Effect of Evolution of Carbon Structure during Torrefaction in Woody Biomass on Thermal Degradation"

_ijerph, 2022, doi:10.3390/ijerph192416831_

Round 1
Reviewer 1 Report
In the manuscript, the structural variations of biomass during torrefaction were also explored, and effect on thermal degradation upgraded by torrefaction were investigated. Besides, the kinetic of the catalytic pyrolysis process was examined. The manuscript was well organized, the obtained results are original and credible. However, the manuscript needs some minor revisions before accepted. Some main questions are as follows:
1. The abstract should show more briefly-clearly what problem you want to reviews and what contribution of the manuscript you want to provide.
2. In the Introduction, the review for bond cleavage should further suppressed and the originality of this work should be more clearly stated after fully reviewing the related previous works.
3. The DTG curve was analyzed by the gaussian curve fitting method, but the introduction was missed. Furthermore, the assumed corresponded reaction for each peak was lack of evidence.
4. The mechanism on torrefaction should be indicated more explicitly and search more evidence.
5. The authors are recommended to state the wider applicability of their results.
6. Please check the language and grammar issues throughout the manuscript.
Author Response
In the manuscript, the structural variations of biomass during torrefaction were also explored, and effect on thermal degradation upgraded by torrefaction were investigated. Besides, the kinetic of the catalytic pyrolysis process was examined. The manuscript was well organized, the obtained results are original and credible. However, the manuscript needs some minor revisions before accepted. Some main questions are as follows:
Issue 1. The abstract should show more briefly-clearly what problem you want to reviews and what contribution of the manuscript you want to provide.
Discussion: We have revised the abstract more briefly-clearly in revised manuscript.
Issue 2. In the Introduction, the review for bond cleavage should further suppressed and the originality of this work should be more clearly stated after fully reviewing the related previous works.
Discussion: Effect of torrefaction on bond energy or thermal degradation is rare. We have added relevant reference on bond cleavage in the section of results and discussion and clearly indicated the originality of this work as “The quantitative information on structural evolution of biomass torrefaction is still insufficient. These structural variations have significant effect on cleavage of bond in biomass during thermal degradation. A comprehensive insight into mechanism on biomass microstructural evolution to predict thermal degradation after torrefaction is also indistinct”.
Issue 3. The DTG curve was analyzed by the gaussian curve fitting method, but the introduction was missed. Furthermore, the assumed corresponded reaction for each peak was lack of evidence.
Discussion: The analyzed method was added in the experimental section. The assumed corresponded reaction for each peak was detailed in reference [32-33], and we consulted it.
Issue 4. The mechanism on torrefaction should be indicated more explicitly and search more evidence.
Discussion: We have indicated more explicitly and consulted more reference.
Issue 5. The authors are recommended to state the wider applicability of their results.
Discussion: We have stated the work in a wider applicability for bioenergy utilization.
Issue 6. Please check the language and grammar issues throughout the manuscript.
Discussion: We have checked the language and grammar issues throughout the manuscript carefully.
Reviewer 2 Report
This work intended to investigate the carbon structure of torrefied woody biomass using various analyses techniques. Although there are many works on the analyses of torrefied biomass in the literatures, this work shows some interesting data, however, the reviewer has some comments before the manuscript is accepted for publication as follows:
1 Please add two more keywords (generally up to 6 is allowed). Metadata including keywords are important in terms of the searchability of the manuscript if published.
2 Please include a Table of Abbreviations/Nomenclatures.
3 Please synchronize the font size used in the manuscript.
4 Too long paragraphs should be avoided.
5 Reference lumping can be observed on some occasions. Please cite references where they exactly belong; this will prevent reference lumping.
6 Consistency in presenting units should be observed. For instance, either use “200-300 ℃” or “200-300 ℃”. In the current version, both with and without space have been used.
7 Latest trends in biomass torrefaction research, as explained in a recent work entitled “Oxidative torrefaction and torrefaction-based biorefining of biomass: A critical review”, etc. should be discussed. Authors can consider including the mentioned work to elaborate on this.
8 The novelty of the present work against the existing literature in this domain should be more effectively highlighted.
Author Response
This work intended to investigate the carbon structure of torrefied woody biomass using various analyses techniques. Although there are many works on the analyses of torrefied biomass in the literatures, this work shows some interesting data, however, the reviewer has some comments before the manuscript is accepted for publication as follows:
Issue 1 Please add two more keywords (generally up to 6 is allowed). Metadata including keywords are important in terms of the searchability of the manuscript if published.
Discussion: We have added “oxygen functional group; DTG” as key words
Issue 2 Please include a Table of Abbreviations/Nomenclatures.
Discussion: We have added a Table of Abbreviations/Nomenclatures
Issue 3 Please the font size used in the manuscript.
Discussion: We have synchronized the front size.
Issue 4 Too long paragraphs should be avoided.
Discussion: We have clarified the paragraphs.
Issue 5 Reference lumping can be observed on some occasions. Please cite references where they exactly belong; this will prevent reference lumping.
Discussion: We have clarified the reference.
Issue 6 Consistency in presenting units should be observed. For instance, either use “200-300 ℃” or “200-300 ℃”. In the current version, both with and without space have been used.
Discussion: We have modified.
Issue 7 Latest trends in biomass torrefaction research, as explained in a recent work entitled “Oxidative torrefaction and torrefaction-based biorefining of biomass: A critical review”, etc. should be discussed. Authors can consider including the mentioned work to elaborate on this.
Discussion: We have consulted and discussed in section of Introduction.
Issue 8 The novelty of the present work against the existing literature in this domain should be more effectively highlighted.
Discussion: We have indicated the originality of this work as “The quantitative information on structural evolution of biomass torrefaction is still insufficient. These structural variations have significant effect on cleavage of bond in biomass during thermal degradation. A comprehensive insight into mechanism on biomass microstructural evolution to predict thermal degradation after torrefaction is also indistinct”.